# Occupational Stress and Mental Health among Anesthetists during the COVID-19 Pandemic

**DOI:** 10.3390/ijerph17218245

**Published:** 2020-11-08

**Authors:** Nicola Magnavita, Paolo Maurizio Soave, Walter Ricciardi, Massimo Antonelli

**Affiliations:** 1Postgraduate School of Occupational Medicine, Università Cattolica del Sacro Cuore, 00168 Rome, Italy; paolomaurizio.soave@policlinicogemelli.it; 2Department of Woman/Child & Public Health, Fondazione Policlinico Universitario Agostino Gemelli IRCCS, 00168 Rome, Italy; walter.ricciardi@unicatt.it; 3Department of Emergency, Anesthesiology and Resuscitation Sciences, Fondazione Policlinico Universitario Agostino Gemelli IRCCS, 00168 Rome, Italy; massimo.antonelli@unicatt.it

**Keywords:** anxiety, depression, emergency, healthcare workers, infectious disease, insomnia, logistic regression, organizational justice, SARS-CoV-2, sleep

## Abstract

Anesthetist-intensivists who treat patients with coronavirus disease 19 (COVID-19) are exposed to significant biological and psychosocial risks. Our study investigated the occupational and health conditions of anesthesiologists in a COVID-19 hub hospital in Latium, Italy. Ninety out of a total of 155 eligible workers (59%; male 48%) participated in the cross-sectional survey. Occupational stress was assessed with the Effort Reward Imbalance (ERI) questionnaire, organizational justice with the Colquitt Scale, insomnia with the Sleep Condition Indicator (SCI), and mental health with the Goldberg Anxiety and Depression Scale (GADS). A considerable percentage of workers (71.1%) reported high work-related stress, with an imbalance between high effort and low rewards. The level of perceived organizational justice was modest. Physical activity and meditation—the behaviors most commonly adopted to increase resilience—decreased. Workers also reported insomnia (36.7%), anxiety (27.8%), and depression (51.1%). The effort made for work was significantly correlated with the presence of depressive symptoms (r = 0.396). Anesthetists need to be in good health in order to ensure optimal care for COVID-19 patients. Their state of health can be improved by providing an increase in individual resources with interventions for better work organization.

## 1. Introduction

Intensivists have played a vital role in the treatment of patients with coronavirus disease 2019 (COVID-19) caused by severe acute respiratory syndrome coronavirus 2 (SARS-CoV-2). When the first cases of the COVID-19 disease were reported in Italy, it was already clear that SARS-CoV-2 could be transmitted from human to human [1]. Equally evident was the need to adopt very different measures from those previously used to safeguard workers [2]. The rapidity with which the pandemic spread severely tested the ability of the health service to respond. Nationally, there was a grave shortage of protective devices, e.g., masks and disinfectants, and mechanical ventilators for patient care. In many hospitals, emergency rooms, in-patient departments for respiratory and infectious diseases, and intensive care units (ICUs) were suddenly inundated with confirmed and suspected COVID-19 cases, and healthcare personnel were swiftly overwhelmed [3]. In early March 2020, in a matter of days, a second COVID-19 hospital was set up in Rome by the Agostino Gemelli Scientific Hospitalization and Care Institutes (IRCCS) Foundation in conjunction with the Latium Region to treat cases of infection from the new coronavirus, and a team of hospital anesthetists was assigned to this new type of patient.

It has been known for some time that anesthetists are exposed to a high level of hematogenous [4,5] and aerogenic [6] occupational biological risk. During epidemics, front-line anesthetists are among the most vulnerable professional healthcare workers (HCWs) on account of infections and mental health problems [7]. During the COVID-19 pandemic, they were exposed to a completely new, and therefore partially unknown, biological risk. SARS-Cov-2 is transmitted mainly by droplets and close contact but transmission by aerosol is also possible. Viral particles of <5 µm can remain suspended in the air for a few hours and travel over long distances, especially if they are attached to atmospheric particulates [8,9]. Intensive care unit (ICU) staff are at higher risk of COVID-19 infection especially during aerosol-generating airway procedures (e.g., tracheal intubation, replacement/removal of the endotracheal/tracheostomy tube, and bronchial fibroscopy) and cardiopulmonary resuscitation (CPR) [10]. The epidemic has forced international bodies to modify guidelines for resuscitation [11,12,13,14] and surgical practices [15,16,17,18]. Anesthetists have to determine the best therapeutic management of patients and keep themselves safe while doing so [19,20]. Moreover, they have to test the effectiveness of these new indications and new therapeutic protocols directly, without being able to rely on evidence from previous experiences, under conditions of extreme pressure when all available beds are allocated. Consequently, the COVID-19 pandemic led to a certain amount of therapeutic and logistic uncertainty that required close monitoring and the presence of trained personnel [21]. Younger anesthesiologists often found themselves experiencing greater uncertainty and more difficulty in performing according to standard procedures [22]. In some cases of COVID-19, not enough importance was given to the need to protect HCWs. As a result, many anesthesiologists were among the front-line physicians who died after contracting the disease while caring for patients [23].

The new working conditions during the pandemic have exacerbated compassion fatigue resulting from an empathic response to suffering people [24,25], which can be associated with therapeutic failure and contact with relatives at the end of life [26,27]. Workplace violence was already a significant risk for emergency medicine physicians [28] that had a five-fold risk of suffering physical violence compared to doctors working in services (odds ratio (OR) = 5.12, confidence interval (95%CI) = 2.21–12.39) [29]. The current pandemic has caused significant mortality over a short time and has necessitated an increase in provision of both critical care and palliative care for anesthesiologists deployed to units caring for patients with COVID-19 [30]. For this reason, anesthetists, and especially female junior doctors specializing in anesthesia, have been found to be at high risk of burnout [31,32,33,34], a syndrome characterized by emotional exhaustion, low personal accomplishment, and depersonalization, and to have a high suicide rate [35]. Given the crucial role they play in managing the epidemic, the realization of online platforms to provide free mental health care for anesthetists has been proposed [36].

All these considerations suggest that anesthesiologists who have had to face up to the epidemic have perceived levels of organizational correctness and stress that have resulted in insomnia, anxiety, and depression. To test this hypothesis, we set out to evaluate the health of anesthetists engaged in the treatment of patients with COVID-19. Our assessment focused particularly on the conditions of occupational stress to which they were exposed at the height of the epidemic, on the perception of correctness in the organization of safety measures, and on the main factors that could increase resilience. As an outcome, we measured the possible early psychological effects of distress. We invited all anesthesiologists of a COVID-19 hospital in Rome to provide anonymous information on these topics via an online platform.

## 2. Materials and Methods

### 2.1. Participants

At the “A. Gemelli” University hospital in Rome, a cross-sectional study was conducted on anesthetists directly engaged in the care of suspected or confirmed cases of COVID-19.

All workers (155) were confidentially contacted by email and invited to participate in the survey proposed on the SurveyMonkey© online platform. The answers were collected anonymously on a specific file without any individual reference. Participation was completely voluntary and no economic incentive was provided for response. Participants were enrolled between 27 April and 27 May 2020. Two weeks after commencing the investigation, a reminder mail was sent with the preliminary results of the study.

Ninety out of the 155 eligible workers completed the survey (participation rate = 58.1%). Participants were mainly young (76.7% under 35 years of age), female (47, 52.2%) workers. Forty (44.4%) had a permanent employment contract, while the remainder had been given short-term internships.

Most workers (66.6%) were living with a partner without the presence of children (84.4%) or relatives who were not self-sufficient (94.4%). A noteworthy feature was the absence of persons who could help in case of need (23.3%). Approximately one in four (23.3%) reported unprotected exposure to SARS-CoV-2 patients (Table 1).

The research was conducted in accordance with the Helsinki Declaration. Ethics approval was obtained from the Catholic University Ethics Committee (ID 3292).

Given the cross-sectional study design, there was no imputation for missing data and the results were based on completed survey responses.

### 2.2. Questionnaire

Before carrying out our study, we consulted a focus group composed of qualified anesthetists in order to identify the specific stress factors affecting the profession and provide indications for compiling the questionnaire. On the basis of this interview, three questions were formulated concerning the workload and specific occupational stressors that arose during the epidemic and the subsequent lifestyle changes. The questionnaire was composed of 59 questions divided into 6 sections. The average time required for completion was 6 min.

The first section (6 items) regarded socio-demographic factors that could influence the outcome, e.g., gender, age class (younger or older than 35 years), marital status (single/coupled), presence of underage children or cohabiting non self-sufficient relatives, and presence of persons who could help in case of need.

The second section (5 items) investigated the main changes in occupation and lifestyle resulting from the epidemic. The anesthesiologists were asked to indicate the extent of their workload in the first months of the epidemic compared to the past, by choosing one of 5 responses ranging from “much less than usual” to “much greater than usual”. Similarly, they were required to indicate (with the same 5-point Likert-type scale) how much time had been spent on physical activity and, respectively, meditation, prayer, or spiritual/mental activities in the first months of the epidemic, compared to the past.

Furthermore, they were asked to indicate whether they agreed that during the epidemic, their work had become more monotonous and repetitive, and that the task of informing relatives of the death of a patient had been more frequent; in these two questions, they could chose the answer from two 5-point Likert scales ranging from “I don’t agree” to “I absolutely agree”.

Occupational stress was assessed with the “Effort Reward Imbalance” (ERI) model [37]. The Italian version [38] of the short questionnaire [39] was used. This consisted of 10 items with responses ranging on a 4-point Likert scale from “1 = strongly disagree” to “4 = strongly agree”. The effort subscale was based on three questions (e.g., “I’m always under pressure for the workload”); the total score ranged from 3 to 12. The reward sub-scale was based on seven questions (e.g., “Considering all my efforts and what I have achieved, I receive the respect and prestige I deserve at work”); consequently, this score ranged from 7 to 28. Internal consistency reliability, which is an indicator of the adhesion of the questions to the same construct, is usually measured with Cronbach’s alpha, a statistic calculated from the pairwise correlations between items, ranging between negative infinity and one [40]. The reliability of the two effort and reward sub-scales in this study was 0.679 and 0.731, respectively, both in the acceptable range [41]. Stress was measured as a weighted relationship between effort and reward: conventionally, an ERI greater than unity is believed to indicate an imbalance between effort and rewards.

Organizational justice, which refers to processes and procedures employed to resolve conflicts or allocate resources [42], was measured using the Italian version [43] of the Colquitt Scale [44,45]. Procedural justice (PJ) was measured by 7 items (e.g., “Were you able to express your views and feelings during those procedures?”); informational justice (IJ) was measured with 5 items (e.g., “Do you think the communications you received were reliable?”). Each question was answered according to a 5-point Likert scale, from 1 = “I absolutely don’t agree” to 5 = “I strongly agree”, thus giving a sub-scale ranging from 7 to 35, and an IJ subscale ranging from 5 to 25. The score for organizational justice (OJ), which is the sum of the two previous sub-scales, ranged from 12 to 60. In this study, the reliability of the questionnaire, measured by Cronbach’s alpha, was 0.861 for PJ (good) and 0.696 for IJ (acceptable).

Sleep quality was measured with the Italian version [46] of the “Sleep Condition Indicator” (SCI) [47], a brief scale (8 items) that evaluates insomnia disorder in everyday clinical practice, according to the Diagnostic Statistic Manual 5 (DSM5). Each question (e.g., “How many nights a week have you had a problem with your sleep during the past month?”) was graded on a 5-point Likert scale. The final score ranged between 0 and 32; a score of ≤16 revealed possible insomnia disorder. Cronbach’s alpha was 0.860 (good).

Psychological symptoms of anxiety (9 items) and depression (9 items) were measured with the Italian version [48] of the “Goldberg Anxiety and Depression Scale” (GADS) [49]. Typical questions were: “Have you felt keyed up, on edge?” for anxiety, and “have you lost confidence in yourself?” for depression. The reference time frame was the last 15 days. The anxiety and depression scores were calculated by adding one point for each positive answer. Persons with an anxiety score of 5 points or more, or a depression score of 2 or more, are over 50% more likely to be diagnosed as anxious or depressed by a psychiatrist; this probability increases rapidly in proportion to the score. In this study, the reliability of the GADS subscales, measured by Cronbach’s alpha, was 0.725 for anxiety and 0.627 for depression (acceptable).

### 2.3. Statistics

Categorical variables were measured in terms of frequency. Continuous variables were analyzed in terms of mean and standard deviation. Student’s *t*-test or Mann–Whitney’s U test for nonparametric variables were used to compare the distribution of continuous variables in subgroups of workers defined by gender or age.

Simple linear regression was used to evaluate the effect of demographic, social, or work-related factors on occupational stress and to assess the effect of stress and justice perception on anxiety and depression scores.

Logistic regression was used to evaluate the relationship between stress, organizational justice, and cases of anxiety or depression.

Analyses were performed using IBM/SPSS 26.0 (IBM Corporation, Armonk, NY, USA).

## 3. Results

### 3.1. Changes due to the Pandemic, Stress and Perceived Justice

During the SARS-CoV-2 pandemic, workers reported an increase or high increase in workload (63.3%) in addition to an increase both in monotony (33.4%) and the need to inform relatives of the death of a patient (48.9%). Free time spent on physical activity and meditation was reduced or severely reduced for most of the participants (84.4% and 57.8% for physical and spiritual activities, respectively) (Table 2).

Accordingly, evaluations of occupational stress indicated a rather high average effort level and a moderate reward level: effort (range 3–12) = 8.17 ± 1.81; reward (range 7–28) = 16.48 ± 3.56. The mean values of the two stress-related variables were, respectively, 68% and 58% of the maximum values of their scales. Consequently, ERI was higher than unit, indicating high work-related stress in 71.1% of anesthesiologists.

Perceived levels of organizational justice reported in the sample were far from optimal. Procedural justice (range 7–35) was, on average, 17.58 ± 5.02, i.e., 50% of the maximum; informative justice (range 5–25) was 13.87 ± 3.39, i.e., 55% of the maximum. Organizational justice was 31.44 ± 7.38, 52% of the maximum. An analysis of the responses indicated that workers perceived a scarce ability to influence and modify the outcome of procedures. This was accompanied by a general uncertainty about the rationality and accuracy of the procedures to be adopted. In addition, workers expressed doubts regarding the reliability of communications and explanations concerning the pandemic in progress.

### 3.2. Prevalence and Distribution of Neuropsychological Disorders

The quality of sleep in the sample was not high. The SCI final score was, on average, 19.96 ± 5.43, 62% of the theoretical maximum. According to the DSM5 criteria, 33 workers (36.7%) suffered from insomnia.

The GADS yielded an average score of 3.20 ± 2.34 for anxiety and 2.10 ± 1.84 for depression. Twenty-five workers (27.8%) were classified as anxious and 46 (51.1%) as depressed.

Female anesthesiologists reported higher effort, lower reward and justice, lower sleep quality, and higher anxiety and depression scores than their male colleagues; however, the disparity failed to reach statistical significance. Similarly, younger workers reported non-significant higher stress levels, lower justice perception, and a higher prevalence of anxiety and depression compared to older colleagues.

In a simple linear regression model, the level of stress perceived by each worker (ERI) was not significantly associated with gender, age, presence of family or social risk factors, nor with changes in the levels of monotony of work and workload. Although compassion fatigue was high in distressed anesthetists, it failed to reach a significant level. Exposure to confirmed COVID-19 patients without adequate protection and the perception of low organizational justice were significantly associated with stress (Table 3).

Simple linear regression analysis showed that anxiety and depression levels in the sample were significantly associated with efforts made at work (Effort) (Table 4).

Univariate logistic regression revealed that effort was a significant predictor of the risk of being depressed (OR = 1.539; 95%CI = 1.162–2.039). This association was confirmed in a multivariate logistic regression model, adjusted for gender, age, rewards, and justice (OR = 1.709; 95%CI = 1.225–2.386) (Table 5).

## 4. Discussion

This cross-sectional study, conducted on a large percentage of the anesthetists employed in a COVID-19 hub hospital in the Latium region of Italy, demonstrated that alongside existing occupational stressors connected with their work, there were also additional stressors closely related to the pandemic. In fact, these physicians perceived organizational problems related to the sphere of procedural and informational justice and were exposed to high levels of occupational stress. They reported sleep problems, anxiety, and depression with a frequency that was not negligible. Effort, i.e., occupational psychophysical commitment, was the major predictor of these problems. The question is: how many of the mental health problems that we have observed are attributable to the epidemic?

In epidemiological studies designed to evaluate the effect of the pandemic on the mental health of HCWs, a problem often arises due to a lack of reference values. Medical activities are stressful even when there is no pandemic, and some doctors suffer from excessive stress, anxiety, and depression for causes other than COVID-19 [50,51,52]. Currently, no longitudinal studies are available to enable us to compare the situation before, during, and after the outbreak of COVID-19. Available cross-sectional studies, which have found rates of anxiety and depression in anesthetists comparable to ours [53,54], lack control groups when external references are used, e.g., stress levels or mental health in other populations; consequently, they are often unreliable due to methodological or chronological differences [55]. To get an idea of the pre-COVID-19 situation, we can consider some of the most recent studies conducted on HCWs before the pandemic. In Australian nurses, in 2018, the prevalence rates of depression, anxiety, and stress were found to be 32.4%, 41.2%, and 41.2% respectively [56]. The overall prevalence of psychological distress among nurses in a teaching hospital in Malaysia in 2019 was 41%, and the prevalence of stress, anxiety, and depression were 14.4%, 39.3%, and 18.8%, respectively [57]. In a 2017 study, Australian midwives reported moderate/severe/extreme levels of depression (17.3%), anxiety (20.4%), and stress (22.1%) symptoms [58]. In the USA, nearly one-quarter of hospice workers were moderately to severely depressed, and nearly one-third reported moderate to severe symptoms of anxiety [59]. In 2018, 58.1% of doctors who worked at cancer hospitals were identified as having burnout, depression (12.3%), and anxiety (19.4%) [60]. Beyond the fact that these data demonstrate the existence of some problems even before the pandemic, due to methodological problems, they are not directly comparable with our findings. To avoid this problem, we have identified, in the literature, an experience conducted on a comparable sample. The study was conducted on the HCWs of a local health unit [61] in the same Italian region, during the same chronological period, with a similar method. In that survey, 86 physicians, whose work did not include unprotected exposure to COVID-19 patients, reported an average stress level measured with the ERI model of 0.90 ± 0.31, which was significantly much lower than that observed in our study (1.25 ± 0.51). Moreover, in that situation, which is the one commonly experienced by National Health Service doctors, 37.2% of the sample had high stress levels (ERI > 1), 9.3% suffered from anxiety (GADS score ≥ 5), and 11.6% suffered from depressive symptoms (GADS score ≥ 2). In our present study, an obvious difference between the aforementioned levels and those of the COVID-19 hospital anesthesiologists was observed. In our opinion, this difference could be attributed to the pandemic.

The results of our survey are not surprising if we analyze data reported in the literature. Excessive occupational stress could result from concerns about the risk of infection and lack of confidence in the safety measures adopted [62]. During epidemics, risk management in hospitals is not easy, inadequate risk assessment can occur [63], and it is not always possible to adequately safeguard the health of HCWs [64]. Anesthesiologists involved in aerosol-generating procedures, such as tracheal intubation, are at an elevated risk of acquiring COVID-19 [65]. A multicenter study conducted in 503 hospitals in 17 different countries observed a 10.7% incidence of infection in anesthetists participating in the tracheal intubation of patients with suspected or confirmed COVID-19, and the risk rate was higher in women [66]. Other medical activities involving patients with COVID-19 have also been associated with unwanted exposures: 11.4% of Chinese anesthetists who performed spinal anesthesia on patients with COVID-19 were infected [67]. In our study, although most unprotected exposures were prevented by adopting safety measures in the hospital, twenty-one of the respondents (23%) reported at least one work-related unprotected exposure. This percentage was significantly lower than that recorded in the United States, where 58% of anesthesiologists and affiliated intensive care providers at the Columbia University Irving Medical Center reported unprotected exposures, 54% of which were high-risk [68]. Nevertheless, our data indicate that further improvements can be made in anesthetist safety levels.

The anesthetists participating in our survey were well aware of the risk and the need to adopt scrupulous safety procedures. However, the novel situation and the difficulty encountered in changing work habits undoubtedly prompted anesthetists to make an additional effort demonstrated in the high levels recorded in the effort subscale of the Siegrist ERI model. Despite the additional effort during this period, doctors received no increase in material rewards for their work. On the contrary, in some cases, the latter decreased due to the unsatisfactory results of some therapies. This created an imbalance in the relationship between effort and rewards, resulting in a generalized condition of excessive occupational stress. This condition is not new in front-line workers and has been reported in all health professionals who have dealt with the epidemic [69]. Compassion fatigue resulting from symptom management and end-of-life care may also have induced increased stress. At the time of our survey, three out of four anesthesiologists were suffering from distress; over prolonged periods, this can seriously affect physical and mental health, lead to an increase in errors, and, ultimately, become dangerous for patients.

Most of the anesthesiologists who participated in the survey also reported a reduction in their physical activity during the pandemic. Physical activity/exercise was the most common and effective form of coping behavior adopted by 59% of American HCWs exposed to the pandemic [70]. A reduction in the time devoted to meditation was also reported in our survey. Spirituality is a well-known resource against work-related stress [71,72]. The simultaneous weakening of two important ways of combatting stress put the mental and physical wellbeing of our sample at high risk.

A considerable number of reports have been published on mental health problems that have arisen in HCWs due to COVID-19. After Lai et al. [73] estimated the prevalence of distress in Chinese HCWs to be 71.5% (a rate comparable to that observed in our study), several reviews [74,75,76,77] and three meta-analyses [78,79,80] have confirmed a high prevalence of psychological distress, poor sleep quality, anxiety, and depression/depressive symptoms in HCWs working with COVID-19 patients. In HCWs treating COVID-19 patients, the pooled prevalence of anxiety has been estimated to range from 23.2% [79] to 32.0% [80] and that of depression from 22.8% [79] to 28% [80]. These values are similar to those observed in our investigation. In the literature, women and young anesthetists generally have a higher prevalence rate for these problems than their colleagues [81,82,83,84]. In our sample, the prevalence of distress (ERI > 1) in female anesthetists was 78.7% and that of anxiety was 29.8%, compared to 62.8% and 25.6%, respectively, in their male colleagues. Given the small sample size, the difference was not significant, but the trend was not rejected. The same was true for the greater frequency of stress, anxiety, and depression in younger workers, who reported these problems in 72.5%, 30.4%, and 55.1%, respectively, as compared to 66.7%, 19.0%, and 38.1% in their older colleagues.

In addition to confirming the existence of this phenomenon, we made a deeper analysis of its causes with a statistical model, demonstrating that the effort made during work is significantly associated with mental problems. This result corroborates findings for another high-stress profession – the police force—in which the Effort score was a predictor of anxiety and depression [85] sick leave [86], metabolic syndrome [87], and sleep problems [88]. This observation clearly indicates that to prevent occupational stress, instead of increasing economic or other rewards, it is more important to reorganize the work, so as to reduce the effort made by each individual worker.

The main advantage of this study is that it was conducted on a representative sample of anesthesiologists from a hub hospital at the height of occupational commitment to the COVID-19 epidemic. Its principal limitation consists in the cross-sectional nature of the study, which prevents us from making inferences on the sequence of events and does not exclude the possibility of reverse causation. In fact, it is well known that persons undergoing anxiety and depression can have sleep disturbances and suffer from occupational stressors more than others [89]. However, the proposed sequence, in which exposure to the working conditions produced by the epidemic caused stress and consequently induced anxiety, depression, and insomnia, seems to us to be the most plausible. Another limitation, common to all the studies conducted in this period, is that they refer to a rather short period, during which the chronic effects of stress have not had time to manifest themselves. Only investigations conducted over a longer period and potential future longitudinal studies will provide us with a better knowledge of the relationships between work factors and alterations in the state of the physical and mental health of workers. Finally, self-reporting represents a further limitation of this study, despite the use of standardized questionnaires, which, in previous research, have proven to be well-correlated to the phenomena investigated.

Looking at the survey results, we were quite surprised at the fairly high rate of colleagues who said they had uncontrolled exposures and we are determined to investigate these aspects for hospital job safety reasons. A repetition of the survey in the stabilization phase of the epidemic could make us understand how occupational stress levels change over time. A repeated cross-sectional approach, moreover, can give some insights on the relationships between stress, sleep, and mental health problems.

## 5. Conclusions

Our analysis of the working conditions in an important health center dedicated to the treatment of patients with COVID-19 has revealed the possibility of improving the condition of the anesthesiologist-intensivists, of counteracting their state of stress, and thereby preventing negative effects on mental health. Interventions to improve worker resilience and deal effectively with COVID-19 stress in HCWs have been proposed [90]. These programs will certainly be more effective if accompanied by organizational measures aimed at reducing the effort of workers in health care activities. Numerous structural measures have proved effective in reducing the psycho-physical burden of workers. Science-based, data-driven cooperative organization measures, including balance between workload and staffing, shift scheduling, workplace environment design, and employee training, are the first defense against workers’ distress [91]. The workers should receive education and training to mitigate fatigue and fatigue-related risks [92] and must closely monitor their sleep health conditions, relaxation abilities, and general well-being [93]. Improving the health conditions of anesthesiologists is also of crucial importance in order to guarantee the quality of care, especially during a pandemic [94].

## Figures and Tables

**Table 1 ijerph-17-08245-t001:** Characteristics of the population.

Variable	N	%
Gender, male	43	47.8
Age, <35 years	69	76.7
Family status, single	30	33.3
With children	14	15.6
With old/disabled relatives	5	5.6
Without social support	21	23.3
Reporting unprotected exposure to COVID-19 patients	21	23.3

**Table 2 ijerph-17-08245-t002:** Changes reported during the COVID-19 outbreak, and prevalence of high stress, insomnia, anxiety, and depression in anesthesiologists.

Reported Effect	N	%
Increased/much increased workload	57	63.3
The work became more repetitive and monotonous	30	33.4
More frequent need to inform of the death of a relative	44	48.9
The time for physical exercise was shorter/much shorter	76	84.4
The time for meditation was shorter/much shorter	52	57.8
High stress (effort/reward weighted ratio >1)	64	71.1
Insomniac (SCI score ≥16)	33	36.7
Anxious (GADS anxiety score ≥5)	25	27.8
Depressed (GADS depression score ≥2)	46	51.1

SCI = Sleep Condition Indicator; GADS = Goldberg Anxiety and Depression Scale.

**Table 3 ijerph-17-08245-t003:** Linear regression analysis. Relationship between socio-demographic factors, work changes, organizational justice and perceived work-related stress (ERI).

Predictors	ERI	
	Standardized Beta	*p*
Gender	0.084	0.407
Age	0.029	0.809
Family status	−0.003	0.973
Children	−0.147	0.238
Relatives or disabled people	0.006	0.954
Without social support	0.143	0.171
Workload	0.120	0.213
Monotony	0.118	0.226
Compassion fatigue	0.190	0.057
Unprotected exposure	−0.248	0.014
Organizational justice	−0.383	0.000
Determination coefficient of the model (R^2^)		0.329

Gender: 0 = male, 1 = female; Age class: 0 = < 35 years; 1 = > 35; Family status: 0 = single, 1 = married; Children: 0 = yes, 1 = no; Relatives or disabled people: 0 = yes, 1 = no; Without social support 0 = yes, 1 = no; Workload: 1 = much less, 5 = much greater; Monotony: 1 = much less, 5 = much greater; Compassion fatigue: 1 = much less, 5 = much greater; Unprotected exposure: 1 = yes, 2 = no; Organizational justice: score range = 12–56.

**Table 4 ijerph-17-08245-t004:** Linear regression analysis. Relationship between demographic factors, stress, justice, and mental health variables.

Variable	Anxiety		Depression	
	Standardized Beta	*p*	Standardized Beta	*p*
Gender	0.085	0.435	0.007	0.943
Age	−0.095	0.386	−0.110	0.282
Effort	0.224	0.050	0.396	0.000
Reward	0.099	0.417	0.141	0.220
Procedural justice	−0.127	0.323	−0.155	0.200
Informational justice	0.004	0.973	−0.059	0.625

Gender: 0 = male, 1 = female; Age class: 0 = < 35 years; 1 = > 35; Effort range 3–12; Reward range 8–25; Procedural justice range 7–35; Informational justice range 5–21.

**Table 5 ijerph-17-08245-t005:** Logistic regression analysis. Univariate and multivariate associations of work-related variables with anxiety and depression cases in anesthesiologists.

Variable	Anxiety		Depression	
	Model I OR (95%CI)	Model II OR (95%CI)	Model I OR (95%CI)	Model II OR (95%CI)
Effort	1.216 (0.925–1.599)	1.308 (0.938–1.823)	1.539 (1.162–2.039) ***	1.709 (1.225–2.386) ***
Reward	1.023 (0.897–1.166)	1.142 (0.961–1.357)	0.989 (0.880–1.112)	1.103 (0.941–1.292)
Procedural justice	0.945 (0.854–1.046)	0.979 (0.865–1.109)	0.965 (0.888–1.050)	0.990 (0.886–1.107)
Informational justice	0.883 (0.769–1.014)	0.870 (0.724–1.045)	0.935 (0.825–1.060)	0.962 (0.812–1.140)

Model I: univariate; Model II: multivariate, adjusted for gender, age, and containing all the work-related variables (effort, reward, procedural justice, and informational justice). *** *p* < 0.001.

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
