# Peer review of "Occupational Stress and Mental Health among Anesthetists during the COVID-19 Pandemic"

_ijerph, 2020, doi:10.3390/ijerph17218245_

Round 1

Reviewer 1 Report

Introduction: still too general, long, and unneccessary description such as  "Anaesthetists have to determine the 55 best therapeutic management of patients and to keep themselves safe while doing so [19, 20]. 56 Moreover, they have to test the effectiveness of these new indications and new therapeutic protocols 57 directly, without being able to rely on evidence from previous experiences, under conditions of 58 extreme pressure when all available beds were allocated. Consequently, the COVID-19 pandemic led 59 to a certain amount of therapeutic and logistic uncertainty that required close monitoring and the 60 presence of trained personnel [21]. Younger anaesthesiologists often found themselves experiencing 61 greater uncertainty and more difficulty in performing according to standard procedures"

it does not show any needs of research for anaesthetiests.

Method: Table 2 showed Changes reported during the COVID-19 outbreak, and prevalence of high stress, insomnia, 185 anxiety, and depression in anaesthesiologists, but it is quite confusing what these are present for. Changes of these prevalence was statistically significant?

I don't see the logic to describe the statistical results in result section.

Discussion: it is hard to follow logic because it was not stated based on the order of results.

Author Response

Introduction: still too general, long, and unneccessary description such as  "Anaesthetists have to determine the 55 best therapeutic management of patients and to keep themselves safe while doing so [19, 20]. 56 Moreover, they have to test the effectiveness of these new indications and new therapeutic protocols 57 directly, without being able to rely on evidence from previous experiences, under conditions of 58 extreme pressure when all available beds were allocated. Consequently, the COVID-19 pandemic led 59 to a certain amount of therapeutic and logistic uncertainty that required close monitoring and the 60 presence of trained personnel [21]. Younger anaesthesiologists often found themselves experiencing 61 greater uncertainty and more difficulty in performing according to standard procedures" it does not show any needs of research for anaesthetiests.

Response: The reviewer is quite right in pointing out that the introduction devotes ample space to the working conditions of anesthesiologists, and consequently is lengthy. In the part that the reviewer cites, we tried to summarize 4 articles published in 2020 on the condition of anesthesiologists during the COVID-19 epidemic. We have not been able to further synthesize these studies, to which we refer for a more complete analysis; however, we are convinced that knowledge of the anesthetists' working conditions is necessary to interpret the results of the study. The idea of our research project was born from the examination of the working conditions of the anesthesiologists.

In the introduction we followed a logical scheme, describing first of all the setting (first paragraph), then the professional risks of the anesthesiologists, in ordinary conditions and during the Covid-19 pandemic (second paragraph). We then dealt with the possible occupational stressors of anesthesiologists (third paragraph) and indicated reasons and objectives for this study (fourth paragraph).

Method: Table 2 showed Changes reported during the COVID-19 outbreak, and prevalence of high stress, insomnia, 185 anxiety, and depression in anaesthesiologists, but it is quite confusing what these are present for. Changes of these prevalence was statistically significant?

Response: The reviewer correctly points out a topic that we have emphasized several times. Our study was cross-sectional, so we asked the participants what, in their opinion, the changes from the past had been. We explained in the Methods section of the manuscript that the request was to indicate the change between the first months of the epidemic compared to the past. The first part of the Table 2 refers to the changes reported by the workers. In the second part of the table we have indicated the results of the cross-sectional analysis of the parameters of sleep, anxiety and depression, measured at the time of the survey. There can be no before-after statistical comparison.

I don't see the logic to describe the statistical results in result section.

Response: We agree with the reviewer. The statistics used are described in the Statistics sub-chapter of the Methods section. The analysis of the results in a study that evaluates so many parameters is complex and the results must be reported in a very orderly way. For this reason, we have divided the results into sub-sections. The statistics follow the sequence of these sections, and are reported in the statistical methods section 2.3 in the same order. In the first sub-section, 3.1, socio-demographic characteristics are analyzed in terms of frequency. In the sub-section 3.2, the percentage of workers who reported a change is reported as frequency. In the same sub-section, moreover, the values of the variables stress and justice are indicated as means and standard deviations. In the sub-section 3.3 we look at the effects on sleep, anxiety and depression. The variables are always processed as averages, while the percentages of subjects exceeding the cut-off are indicated as frequencies. In lines 205-210 we made some comparisons of these frequencies, with the chi-square test, without significant results.

After the description of the population, we investigated the causes of stress, anxiety and depression by simple linear regression. In these models we placed stress (Table 3), anxiety and depression (Table 4), respectively, as the continuous dependent variable.

Given the association between stress and mental health, we investigated, through logistic regression, the relationship between the variables that indicate stress (effort and reward) and justice (procedural and informative) with the presence of cases of anxiety and depression (Table 5). In this case, the dependent variable was being classified as anxious or depressed, respectively.

Discussion: it is hard to follow logic because it was not stated based on the order of results.

Response: In the discussion, we have summarized the results that originate from the complex series of statistical analyzes, placing those that seem most relevant to us in the first part. We then compared the results of this survey with the data of the international literature. We then exposed the article's strengths and weaknesses and suggested further research developments before drawing conclusions.

Reviewer 2 Report

The study is extremely current, interesting and utilizes relevant questionnaires in data gathering. The version I received already had yellow markings, suggesting that this ms has already gone through at least one circle of referee comments. Therefore my comments are presented as prompt as possible and hopefully these further edit suggestions can strengthen the paper prior publishing.

Abstract:

“The effort made for work was significantly correlated with the presence of depressive symptom” – please give values.

Add mention that this is a cross-sectional study.

Introduction:

“In early March 2020, in a matter of days, a second COVID-19 Hospital was set up in Rome…” – I feel that this sentence is in a wrong place. It could better fit to the “participants” –section. In addition, please explain what “was set up” means. Are the workers the same – are they evaluating their work load in the same workplace, just with different patient group? Or have they experienced also the chance of a workplace – it’s a stressor itself.

Material and methods:

“The reliability of the two effort and reward sub-scales in this study was 0.679 and 0.731, respectively”, give the type of test and reference – (may be clear to experienced researchers but not for juniors)

“In this study the reliability of the questionnaire measured by Cronbach’s alpha, was 0.861 for PJ, 0.696 for IJ.” – reference please.

“Cronbach’s alpha was 0.860” – same comment.

All these should be under the “statistics” –section. In addition, that section needs statistical references.

Results:

Table 1, “Without helping people” is unclear, this could also mean that they don’t have to help anyone while off-duty. Please rephrase.

Tables would be more clear if the % were given first, and then the n in parentheses.

Table 5, “other work-related variables”, please in detail, so the reader does not have to look up for them/guess.

Discussion:

“Pilot study” could be changed to “in this cross-sectional study.

It would be more clear if the main results were summarized in the beginning of discussion. Now the reader has to search for them.

“we have chosen a comparable external sample.” This is a weird place to bring this up, if this is in your study aims, it should be placed differently. If it is just a discussion point, please rephrase. In this current form it is more like a study result, as specific values are given.

“Excessive occupational stress could result from concern about the risk of infections and lack of confidence in the safety measures adopted. During epidemics, risk management in hospitals is not easy, and it is not always possible to adequately safeguard the health of HCWs. Anaesthesiologists involved in aerosol-generating procedures such as tracheal intubation are at elevated risk of acquiring COVID-19.” – please add references

“In our experience, although most unprotected exposures were prevented by adopting safety measures in the hospital…” – please revise, “experience” based knowledge should be avoided, use literature or rephrase.

“The physicians interviewed in our survey” – This is unclear – interviewed? These type of results are not presented.

Overall the discussion is quite jumping from one topic to another and could be clearer, identifying the main results and discussion about them in the same order would be beneficial.

Author Response

The study is extremely current, interesting and utilizes relevant questionnaires in data gathering. The version I received already had yellow markings, suggesting that this ms has already gone through at least one circle of referee comments. Therefore my comments are presented as prompt as possible and hopefully these further edit suggestions can strengthen the paper prior publishing.

Abstract: “The effort made for work was significantly correlated with the presence of depressive symptom” – please give values.

Add mention that this is a cross-sectional study.

Response: We thank the reviewer for appreciating our work. We had already introduced some modifications, now we gladly accept to further modify the text. We added the term "cross-sectional" before survey in the abstract and indicated the correlation values in the abstract.

Introduction: “In early March 2020, in a matter of days, a second COVID-19 Hospital was set up in Rome…” – I feel that this sentence is in a wrong place. It could better fit to the “participants” –section. In addition, please explain what “was set up” means. Are the workers the same – are they evaluating their work load in the same workplace, just with different patient group? Or have they experienced also the chance of a workplace – it’s a stressor itself.

Response: The first paragraph of the Introduction is dedicated to the description of the setting in which the survey was carried out. A section of the hospital complex was dedicated to patients with Covid-19. The staff were all already on duty in the hospital. For this reason it was possible to ask the participants to compare the experience during the first months of the epidemic with the past. To clarify this point, we added “A team of hospital anesthetists was assigned to this new type of patient”.

Material and methods: “The reliability of the two effort and reward sub-scales in this study was 0.679 and 0.731, respectively”, give the type of test and reference – (may be clear to experienced researchers but not for juniors). “In this study the reliability of the questionnaire measured by Cronbach’s alpha, was 0.861 for PJ, 0.696 for IJ.” – reference please. “Cronbach’s alpha was 0.860” – same comment. All these should be under the “statistics” –section. In addition, that section needs statistical references.

Response: We have added a brief description of the alpha coefficient and its interpretation. We then indicated whether the resulting value of the reliability test was acceptable, good, or excellent, as appropriate.

Results: Table 1, “Without helping people” is unclear, this could also mean that they don’t have to help anyone while off-duty. Please rephrase. Tables would be more clear if the % were given first, and then the n in parentheses.

Response: Thanks for the indication. We have replaced in the table " Without persons who could help in case of need”

Table 5, “other work-related variables”, please in detail, so the reader does not have to look up for them/guess.

R.: Thanks for this report. To avoid misunderstandings, we have specified in the legend that the logistic regression model II is multivariate, that is, it contains gender, age and all the variables work-related, effort, reward, procedural justice, informational justice.

Discussion: “Pilot study” could be changed to “in this cross-sectional study.

R.: We agree, we have changed as suggested.

It would be more clear if the main results were summarized in the beginning of discussion. Now the reader has to search for them.

R.: The first paragraph of the Discussion summarizes the main findings of the study. Anaesthetists “perceived organizational problems related to the sphere of procedural and informational justice and were exposed to high levels of occupational stress. They reported sleep problems, anxiety, and depression”. Given the complexity of the study, the discussion of the results is also quite complex. The first paragraph ends with a first question: how many of the mental health problems we have observed are attributable to the epidemic?

In the second paragraph, we try to answer this question. We have identified in the literature some studies that report in anesthesiologists who work with COVID-19 patients rates of stress, anxiety and depression similar to those of our experience, and a study conducted on doctors in a hospital, who instead have a very low rate of problems.

In our judgment, the difference between the high rate of problems in anesthesiologists and the lower rate normally seen in doctors can be attributed to the pandemic. According to literature, one of the leading cause of discomfort may be associated with concern about the risk of infections and lack of confidence in the safety measures adopted. In the third paragraph we deal with this topic.

In the following paragraphs we always compare the results of our study with the literature. The effort made by anesthesiologists for the new ways of working with Covid-19 patients and the consequent occupational stress are treated in the fourth paragraph.

The resilience and anti-stress measures used by anesthesiologists are covered in the fifth paragraph.

The consequences of high levels of stress, anxiety and depression are then treated, with reference to the literature concerning anesthesiologists (sixth paragraph) and other professions (seventh paragraph).

The last paragraph discusses the strengths and weaknesses of the study and indicates the prospects for continuing studies.

“we have chosen a comparable external sample.” This is a weird place to bring this up, if this is in your study aims, it should be placed differently. If it is just a discussion point, please rephrase. In this current form it is more like a study result, as specific values are given.

R.: We absolutely agree. We have reformulated the sentence as follows: “we have identified in the literature an experience conducted on a comparable sample”.

“Excessive occupational stress could result from concern about the risk of infections and lack of confidence in the safety measures adopted. During epidemics, risk management in hospitals is not easy, and it is not always possible to adequately safeguard the health of HCWs. Anaesthesiologists involved in aerosol-generating procedures such as tracheal intubation are at elevated risk of acquiring COVID-19.” – please add references

  1. We added the references in all the points indicated by the reviewer.

“In our experience, although most unprotected exposures were prevented by adopting safety measures in the hospital…” – please revise, “experience” based knowledge should be avoided, use literature or rephrase.

R.: We rephrased: “In our study”.

“The physicians interviewed in our survey” – This is unclear – interviewed? These type of results are not presented.

R.: We changed into: The anaesthetists participating…

Overall the discussion is quite jumping from one topic to another and could be clearer, identifying the main results and discussion about them in the same order would be beneficial.

R.: We have tried to illustrate the logical steps of the discussion.

Reviewer 3 Report

The paper entitled “Occupational stress and mental helth among anesthetists during the CoVID-19 pandemic” makes a good description of the work and mental health aspects that a group of anesthetists faced during a period of the Covid-19 pandemic, but it doesn’t show much important results in the regression analyzes, probably because it was not possible to adequately establish the hypotheses to be tested or not all relevant variables were included, which should be incorporated into the study limitations.

The following aspects should be improved or clarified:

Methodology:

What do the authors mean by “case-control study”?

Were informed consents considered in the online survey?

Include the extreme points of the Likert scale used for the participants to answer the “Effort Reward Imbalance” model (0 to 3?).

Indicate that the alphas of several of the subscales were only close to the cut-off value considered acceptable.

Could the date of application of the survey have influenced the results? How much time back did the questions consider? Did you ask if any of the participants contracted Covid?

Did you ask about the causes of unprotected exposures? This information could have been used in the analyzes or it can at least enrich the discussion of results.

In the discussion section some results are discussed in which no significant differences were obtained (e.g. gender and age), this should be reviewed and interpreted in the sense of suggesting possible causes of not having found differences.

The authors suggest “to reduce the effort made by each individual worker”. Authors should be more specific in their recommendation indicating what measures or actions can contribute to reducing effort during a pandemic of this magnitude.

Author Response

The paper entitled “Occupational stress and mental helth among anesthetists during the CoVID-19 pandemic” makes a good description of the work and mental health aspects that a group of anesthetists faced during a period of the Covid-19 pandemic, but it doesn’t show much important results in the regression analyzes, probably because it was not possible to adequately establish the hypotheses to be tested or not all relevant variables were included, which should be incorporated into the study limitations.

Response: We thank the reviewer for appreciating our work and we agree that the regression analysis did not provide the results we would have expected. In fact, neither sociodemographic variables nor most of the other variables showed a significant association with outcomes (anxiety and depression). It could be that this is due to the small size of the sample, which still included the majority of the population. The variables investigated were chosen on the basis of literature studies, which highlighted an increased risk of occupational stress in frontline staff. Workers mainly expressed concerns about “living problems”, "burden of taking care of patients", "worries about social isolation", "worrying about infecting family members and friends" and "worrying about being separated from family after being infected" [Feng MC, et al. Exploring the Stress, Psychological Distress, and Stress-relief Strategies of Taiwan Nursing Staffs Facing the Global Outbreak of COVID-19 Hu Li Za Zhi. 2020;67(3):64-74. doi: 10.6224/JN.202006_67(3).09]. Consequently, we included these variables as risk factors in our study.

The following aspects should be improved or clarified: Methodology: What do the authors mean by “case-control study”?

Response: Sorry, this was a typo. Our study was a cross-sectional survey.

Were informed consents considered in the online survey?

R.: The letter with which the workers were contacted contained an informed consent form, which they signed and sent to one of the authors (PMS) working in the same department. The form to be completed online began by remembering that participation was free and required the consent of the worker. The procedure of the study was submitted to the scrutiny of the Ethics Committee of the Catholic University of the Sacred Heart and of the Policlinico Gemelli IRCCS Foundation and was approved.

Include the extreme points of the Likert scale used for the participants to answer the “Effort Reward Imbalance” model (0 to 3?).

Indicate that the alphas of several of the subscales were only close to the cut-off value considered acceptable.

Response: we have added the extreme points of the Likert scale and a brief description of the alpha coefficient and its interpretation, specifying when it was excellent, good, or only acceptable.

Could the date of application of the survey have influenced the results? How much time back did the questions consider? Did you ask if any of the participants contracted Covid?

Response: None of the participants contracted Covid-19. The symptom questionnaires refer to the month prior to the survey (for sleep) or the past 15 days (anxiety and depression). We have added this information in the description of the questionnaires. The overall duration of the survey was one month. As a result, the response date is not likely to have significantly affected the results.

Did you ask about the causes of unprotected exposures? This information could have been used in the analyzes or it can at least enrich the discussion of results.

The reviewer's suggestion touches on a very important topic. We were quite surprised at the fairly high rate of colleagues who said they had uncontrolled exposures and we are determined to investigate these aspects for hospital job safety reasons. Unfortunately, online research is very short (5-7 minutes to complete the questionnaire) and does not allow for further investigation. Starting from the indication of the reviewer, we have specified the lines of development of our study in the final part of the discussion

In the discussion section some results are discussed in which no significant differences were obtained (e.g. gender and age), this should be reviewed and interpreted in the sense of suggesting possible causes of not having found differences.

Response: We agree. The sentence was described as follows: In the literature, women and young anaesthetists generally have a higher prevalence rate for these problems than their colleagues [70-73]. We could not find a significant difference, probably due to the small number of observations; however also in this sample young people and women had high levels of stress, anxiety, depression, and sleep disturbances.

The authors suggest “to reduce the effort made by each individual worker”. Authors should be more specific in their recommendation indicating what measures or actions can contribute to reducing effort during a pandemic of this magnitude.

Response: we are grateful to the reviewer because his observation allowed us to enrich the conclusions with some considerations that better illustrate the necessary interventions.

This manuscript is a resubmission of an earlier submission. The following is a list of the peer review reports and author responses from that submission.

Round 1

Reviewer 1 Report

1. This article is uncreative knowledge.
2. Since the COVID-19 outbreak, many articles on this issue have been published, and still, numerous related researches are ongoing.

3. No new or creative knowledge was found in this article. For example, the author didn’t point out the change of depression and anxiety before and after the outbreak of COVID-19.

Reviewer 2 Report

Thank you for giving me the opportunity to review your paper. Here are some suggestions for your paper to be improved.

  1. Introduction: These should be succinct and concise to explain why anesthetists' mental health problems should be studied. Also, all variables what you chose for this study should be explained why these should be studied. Especially, study purpose should be stated clearly.
  2. Methodology section
    1. There should be explanation how you seperated case group and control group. The numbers should be stated even though this study got waiver of IRB. Moreover, what is the evidence of waiver of IRB? This study is not secondary data analysis, so that it should be conducted with IRB approval (There should be defense of this statement).
    2. What is the evidence of deciding sample size?

    3. Specific explanation about instruments should be included with seperated numbers. Also, it should include the reliability and validity of the instrument in the original study. The sub-scale should be explained. It should be stated that whether you received permission or bought the instrument or not. When you revised the original instrument or developed the questionnaire based on the model, there should be result of factor analysis.

  3. Result: there should be more clear connection between research purpose and the result section.  Major findings should be stated in the text.
  4. Discussion: It would be better that your findings should be stated first, and you need to compare these with the previous findings. Paragraph should be organized according to your major findings. 
  5. Overall: Proper numbering is required for each section, and formating revision such as numbering is required for each instrument, ethical consideration, data collection, required sample size, etc.. Also, it is needed to divide paragraphs according to content carefully.

Thank you.

Reviewer 3 Report

Thank you for the opportunity to review this article. The topic is highly topical and needs to be studied further. Below are a number of recommendations that I believe need to be reflected upon and, in some cases, adapted or improved. Thank you very much for your attention.

TITLE

It's got 19 words. It's true that it gives a lot of information but it's long. If you don't find another shorter title you could keep it as it is. Perhaps instead of "of anesthetists" it would be more convenient to put "among anesthetists". 

ABSTRACT

It would be interesting to note the age and percentage of women or men in the sample, as well as the name of the measuring instrument.

KEYWORDS

Very appropriate. For internal coherence, it would be interesting if the words were in alphabetical order and at least one of the words referred to the statistical design.

INTRODUCTION

Lines 28-40. The first paragraph of the introduction describes the situation of COVID-19 and how the first aspects of the spread of the virus in society have developed. However, references from official bodies such as the Ministry of Health or the World Health Organization are missing to support the description.

The rest of the information provided in the introduction provides interesting references.

Again, the last paragraph would need to be revised. Specifically, from line 89 to line 91 it is commented that given the information provided a hypothesis is presented. However, explained in this way, it seems that the starting hypothesis of this study comes from all the studies mentioned above. It would be interesting to put the references for the specific studies that have given rise to this hypothesis. In addition, it is recommended that this paragraph makes clear what the general objective of the study is and what the hypothesis related to it is. Finally, it is interesting to reflect on whether it would be necessary to define several specific objectives and hypotheses or if, on the contrary, one objective and one hypothesis is sufficient. 

MATERIALS AND METHODS

Section 2.1 could be called Participants. In this section, there should be more information about their socio-demographic characteristics: gender, age, marital status, etc.

An example of a GADS scale item is recommended.

RESULTS

The first results could be better fitted in the Participants section.

Table 1 shows the age variable. Why for under 35? If the variable is continuous it would be better to show the mean and standard deviation and not the number and percentage. 

It would be interesting to analyze the relationship between the variables using a correlation coefficient. 

Anxiety and depression play a leading role in statistical analysis, yet the introduction does not define a research objective that makes these variables explicit. 

DISCUSSION

Line 263. If the literature supports the results found, to which literature do you refer? What studies support this statement?

The references set out in the discussion do not start from the introduction. There is no link between the introduction (the theoretical framework, the theoretical foundation) and the discussion of the data, so the data are not being discussed with a previous knowledge base. It is necessary to use references used in the introduction (at least a large majority).

The information that is narrated is interesting and diverse, but it lacks a link to the theoretical framework.

Line 314. Space is missing between the point and the next sentence.

The advantages, limitations, and future lines of research have been adequately addressed. 

REFERENCES

Review the magazine's rules. For example, the month is not usually put, the volume is in italics, instead of a colon before the pages a comma is used, etc.

There are numerous current references. 

I hope the information is useful. Thank you for your time and work.